# Magnesium and the Brain: A Focus on Neuroinflammation and Neurodegeneration

**DOI:** 10.3390/ijms24010223

**Published:** 2022-12-23

**Authors:** Jeanette A. M. Maier, Laura Locatelli, Giorgia Fedele, Alessandra Cazzaniga, André Mazur

**Affiliations:** 1Department of Biomedical and Clinical Sciences, Università di Milano, 20157 Milano, Italy; 2UNH—Unité de Nutrition Humaine, Université Clermont Auvergne, INRAE, F-63000 Clermont-Ferrand, France

**Keywords:** magnesium, brain, neuroinflammation, blood–brain barrier, neurodegeneration

## Abstract

Magnesium (Mg) is involved in the regulation of metabolism and in the maintenance of the homeostasis of all the tissues, including the brain, where it harmonizes nerve signal transmission and preserves the integrity of the blood–brain barrier. Mg deficiency contributes to systemic low-grade inflammation, the common denominator of most diseases. In particular, neuroinflammation is the hallmark of neurodegenerative disorders. Starting from a rapid overview on the role of magnesium in the brain, this narrative review provides evidences linking the derangement of magnesium balance with multiple sclerosis, Alzheimer’s, and Parkinson’s diseases.

## 1. Introduction

Because it efficiently coordinates six oxygen atoms in the first coordination sphere, magnesium (Mg) was essential in the initial chemical processes resulting in the origin and evolution of life. Consequently, Mg is vital in all living systems [1], and its concentrations are tightly and sometimes specifically regulated in tissues and cells. As an example, in the brain cytosolic, Mg is lower than in the skeletal muscle, while in the cerebrospinal fluid, it is more elevated than in plasma [2].

To review the relation between Mg in brain health and disease, a search was completed using PubMed, MEDLINE, Google Scholar from 2000 to date.

## 2. Magnesium in the Brain

In the brain, beyond its role as a metabolite and its involvement in all biochemical pathways [3], Mg is fundamental for nerve signal transmission and the maintenance of ionic homeostasis (Figure 1). Since calcium (Ca) plays an important role in the control of synaptic activity and memory formation, it is relevant that Mg is considered a natural Ca antagonist. Mg controls Ca influx by regulating Ca voltage-dependent channels, while intracellularly, it inhibits Ca release from cytosolic stores through inositol 1,4,5-trisphosphate and ryanodine receptors [4]. Moreover, Mg affects the major excitatory and inhibitory neurotransmission pathways. It is an agonist of the ionotropic gamma aminobutyric acid type A receptor (GABA_A_-R) [5], known to mediate the anxiolytic and hypnotic actions of benzodiazepines [6]. In early development, GABA_A_-R signaling prompts Mg release from the mitochondria, and the increase of cytoplasmic Mg activates mTOR, which facilitates the formation of neural networks [7] through ribosome biogenesis [8]. Mg inhibits the glutamate N-methyl-D-aspartate receptor (NMDA-R) at the physiological membrane potential, which is around −70 mV, when glutamate only acts on the α-amino-3-hydroxyl-5-methyl-4-isoxazole-propionate (AMPA) receptor, thereby preventing sustained stimulation of NMDA-R, which leads to neuronal death [8]. The protective action of Mg is also due to its ability to block the opening of the mitochondrial permeability transition pore and the subsequent release of cytochrome c, which culminates in apoptosis [8]. Mg also protects the integrity and function of the blood–brain barrier (BBB) [9], the impenetrable cellular scaffold that regulates central nervous system (CNS) homeostasis, and therefore, maintains healthy neurological function [10]. The BBB also protects the brain against toxins and pathogens and bidirectionally controls the passage of molecules between the blood and the CNS. It is also a relevant source of various neurotrophins, among which is brain-derived neurotrophic factor (BDNF), which contributes to neuronal plasticity and is essential for learning and memory [11]. It is reported that Mg supplementation augments serum BDNF in patients with depression [12], and the addition of Mg pidolate (5 mM) to a co-culture model of BBB/cerebral organoids increases the expression of BDNF [13]. Focusing on memory, increasing brain Mg through diet enhances neuronal plasticity and long-term memory in young and aged rodents [14] and in Drosophila [15] through different mechanisms.

Mg in the brain also has a role in counteracting oxidative stress and inhibiting the release of vasoactive molecules, among which is substance P (SP) [2]. Moreover, Mg administration is useful in attenuating spasms and autonomic instability in tetanus, eventually acting as an anti-toxin [16].

Not surprisingly, a deficit of Mg in the brain unbalances neural function and has been linked to neuroinflammation and neurodegeneration.

## 3. Magnesium and Neuroinflammation

Mg deficiency is also linked to chronic low-grade inflammation in the brain [4]. Neuroinflammation shares common primary features with peripheral inflammation, such as the activation of resident macrophages, i.e., microglia, the increase of inflammatory mediators, the recruitment of peripheral immune cells, and local tissue injury [17]. Recently, it was shown that Mg deficiency induces neuroinflammation in the mouse brain by upregulating the expression of neuroinflammation-associated genes in the hippocampus and cortex [18]. Accordingly, estrogen decline in ovariectomized rats causes extracellular and intracellular Mg deficiency, which in turn engenders neuroinflammation by upregulating inflammatory cytokines and by activating microglia, events that are mitigated by the administration of Mg [19]. Moreover, in primary microglia, Mg inhibits lipopolysaccharides (LPS)-induced activation of nuclear factor kappa B (NF-κB), the master regulator of inflammation [20].

### 3.1. Magnesium and Inflammatory Mediators in the Brain

The chronic elevation of the levels of inflammatory mediators, among which are cytokines, nitric oxide (NO), prostanoids, and SP, profoundly affects the CNS [4]. In the brain, these molecules are released by the cellular components of the BBB and by cerebral cells, or they are delivered to the CNS through afferent nerves.

SP, the most abundant tachykinin in the brain, is mainly secreted by neurons and acts after binding its receptor, the G-protein-coupled neurokinin 1 receptor, which is highly expressed by neurons and glial cells [21]. SP acts as a neurotransmitter and a modulator of pain perception, mediates the cross-talk between neurons and immune cells, and promotes inflammation [22]. SP is an early contributor to Mg deficiency-dependent neuroinflammation. Low Mg inhibits the principal SP-degrading enzyme, neprilysin, and consequently, SP increase precedes the changes in redox balance and the enhanced production of NO, typically associated with Mg deficit [23]. SP activates the microglia to release cytokines, prostaglandin E2 (PGE2), NO, and reactive oxygen species (ROS), which maintain and amplify the inflammatory response [24]. SP also promotes BBB hyperpermeability, thus unbalancing the brain microenvironment. Consequently, it is implicated in vasogenic edema [25] because in the very early phases of traumatic brain injuries and in various pathologic processes, SP induces cerebral endothelial cells to overexpress caveolin-1, the main component of caveolae, thus increasing protein transport from the blood to the brain by transcytosis. Because of the consequent osmotic gradient, water enters the brain through acquaporin-4 expressed by the perivascular astrocytic processes. Mg administration downregulates aquaporin-4 and reduces the severity of cerebral edema in a murine experimental model [26]. SP also contributes to tight junction opening [25], partly through the induction of tumor necrosis factor (TNF)α and angiopoietin-2, which induce the redistribution zonula occludens-1 and claudin-5, resulting in BBB impairment [27].

NO, a ubiquitous gaseous cellular messenger, has pleiotropic effects in the brain, where it is synthesized mainly by neuronal nitric oxide synthases (nNOS) [28]. Additionally, brain endothelial cells synthesize NO through endothelial nitric oxide synthases (eNOS), primarily influencing resting vascular tone but also modulating neuronal signaling and protecting the CNS from ischemic injury [29]. Accordingly, in a model of vascular dementia, the knock-out of eNOS exacerbates cognitive impairment and brain damage [30]. NO modulates neurotransmission and neuronal metabolism and regulates learning, memory, and sleep, among others [31]. Its overproduction is detrimental, as NO post-translationally modifies many proteins by S-nitrosylation, which has a role in glutamate excitotoxicity; impairs mitochondrial function; reacts with superoxide to generate peroxynitrite, which damages lipids, proteins, and nucleic acids; and increases BBB permeability [32]. Under low Mg conditions, NO is overproduced [33] and damages the brain [34]. Accordingly, elevating Mg in the cerebrospinal fluid is neuroprotective through the inhibition of NO production [35]. Further, Mg inhibits high NOS activity in cortical neurons after ischemia [36]. In a rat model, Mg prevents maternal inflammation-induced fetal brain injury partly by inhibiting nNOS [37].

Prostaglandins (PG), lipid mediators derived from arachidonic acid, are local regulators of brain functions, such as synaptic plasticity. The levels of PGE2 and PGD2 in the brain are maintained at appropriate low levels also through the clearance pathway of the BBB [38]. PG are massively produced in inflammation by neurons and glial cells and released into the cerebrospinal fluid. PG derange the neuronal network and neuro-glial interactions and impairs microglial function [39]. The most abundant prostaglandin in the brain is PGD2, which is rapidly dehydrated to the biologically active PGJ2. In addition to generating and maintaining inflammation, PGJ2 also impairs mitochondrial function, resulting in oxidative stress and apoptosis [40]. It has been proposed that one of the neuroprotective actions of Mg after brain injury might be related to its inhibitory effect on PG synthesis [20].

Because of its high oxidative metabolism that consumes approximately 20% of the total basal oxygen, the brain produces ROS [41]. At low concentrations, ROS act as signaling molecules and plays a role in controlling the growth of neural stem cells, axonal outgrowth and regeneration, and intracellular Ca concentrations. In response to tissue damage or pathogens being activated, microglia overproduce ROS [17], which plays a role in brain aging and neurodegeneration. Mg mitigates the production of ROS in various tissues, including the CNS. In the brain of different experimental models, Mg has been shown to contrast oxidative damage after hypoxia [42], counter maternal inflammation-induced oxidative stress [37], and protect against carbon monoxide-induced brain damage [43].

Because low Mg concentrations activate NF-kB [4,44], it is not surprising that Mg deficiency is associated with increased levels of pro-inflammatory cytokines, pleiotropic signaling proteins participating in various vital processes (Figure 2). Over the past decades, it emerged that cytokines serve a wide array of physiological functions in the CNS, where neurons, microglia, and astrocytes release TNFα, interleukins (IL)-1, and γ-interferon, all controlling synaptic transmission and playing a role in learning and memory [45]. In contrast, elevated levels of cytokines are detrimental and lead to neuronal death and cognitive impairment. Moreover, high amounts of cytokines generate a self-sustaining loop because they perpetuate the activation of NF-κB. Cytokines released by activated microglia induce neurotoxic astrocyte reactivity, which promotes oligodendrocyte and neuronal cell death [46]. In LPS-stimulated BV2 microglial cells, Mg reduces LPS-induced pro-inflammatory cytokines, including TNF-α, IL-1α, IL-1β, and IL-6, promotes the macrophage M2 polarization, and mitigates neuroinflammation in LPS-injected mice [20,47]. In humans, MgSO_4_ treatment significantly reduces the levels of cytokines in the peripheral blood through the inhibition of NF-κB activation and nuclear localization, events mediated by the increased constitutive levels of its inhibitor IĸBα [44]. It is relevant that the administration of Mg constrains inflammation and reduces the risk of cerebral palsy and major motor dysfunction in preterm neonates [48]. In neonate mice exposed to hypoxia/ischemia, a single dose of Mg reduces inflammation and is neuroprotective by transiently modulating gene expression and affecting mitochondrial function [49].

### 3.2. Magnesium and Other Contributors to Neuroinflammation

Mg is a well-characterized Ca antagonist [50], with obvious consequences on neurotransmission and cell function. In the microglia, Ca signaling is mediated by the activation of many metabotropic receptors or store-operated Ca channels, and changes in the intracellular Ca concentration are required for these cells to execute their sensor and effector functions [51]. The increase of intracellular Ca has a role in stimulating mitochondrial ROS production and inducing the release of cytokines. It is by antagonizing Ca entry through the purinergic channels that Mg curtails microglial acquisition of a neurotoxic phenotype after exposure to LPS [51,52]. Mg also regulates the voltage dependence of NMDA-R, which is essential in synaptic plasticity and in the integration of synaptic activity with neuronal activity [53]. While the reduced function of NMDA-R is linked to cognitive impairment, its overactivation results in excitotoxicity [54]. When extracellular Mg is reduced, NMDA-R is overactivated by Ca with consequent hyper-excitation, leading neurons to excitotoxic cell death [2]. In cultured primary neurons, low extracellular Mg causes fluctuations of intracellular Ca, electrographic epileptiform events, mitochondrial depolarization, and, in the end, NMDA-R-dependent cell death [55]. NMDA-R activation also stimulates the release of SP and induces neuroendocrine changes, events which determine oxidative stress and inflammation [4]. Accordingly, the administration of Mg exerts neuroprotective effects in various animal models after cerebral ischemia or trauma. This beneficial action is partly due to the fact that Mg protects against metabolic failure caused by excitotoxic glutamate exposure [56]. It has also been shown that Mg traverses the NMDA-R channel pore and activates the cAMP-response-element binding protein (CREB) in neurons [57]. CREB is a transcription factor that modulates the expression of many genes affecting the function of the human brain, principally those involved in the production of dopamine [12]. Of interest, it has recently been shown that high Mg reduces the expression of NMDA-R in cerebral organoids, miniature structures mirroring the brain [13]. As mentioned before, in this experimental model, high Mg also upregulates BDNF, a neurotrophin with a widespread distribution in the CNS; it is involved in the growth, maturation, and maintenance of neurons and synapses [11]. Consistently, in an experimental model of depression in rodents, Mg exerted anti-depressant activity through the BDNF pathway [58]. A recent randomized clinical trial showed that supplementing Mg and vitamin D increased the circulating levels of BDNF and reduced serum TNFα and IL-6 in obese women. These findings correlate with the positive effect on mood disorders [59]. It is noteworthy that inflammation inhibits the expression of BDNF and its receptor tropomyosin receptor kinase B (TrkB) [60]. Therefore, the increase of BDNF reported in [59] might be due to the reduction of neuroinflammation.

Another feature associated with neuroinflammation is the impairment of the BBB, which not only allows the transport of plasma proteins and immune cells to the brain, but also amplifies neuroinflammation because the endothelial cells and the reactive astrocytes of the BBB release inflammatory mediators [61]. It is known that Mg prevents BBB hyperpermeability induced by the plasma of women with preeclampsia [62], restores BBB permeability after traumatic brain injury or ischemia in rodents [63,64], and in in vitro models of the BBB, protects against LPS-induced damage [9].

Since alterations of Mg levels might favor SARS-CoV-2 dissemination and further aggravate neurological signs and symptoms, it has recently been proposed that derangements of Mg levels might be involved in neuro-COVID and its clinical manifestation [65]. 

## 4. Magnesium and Neurodegeneration

Neurodegenerative diseases, such as Alzheimer’s disease, Parkinson’s disease, and multiple sclerosis, are among the most prevalent disorders with a high burden to the patients, their families, and society. They share a common denominator, i.e., neuroinflammation, which has a relevant role in their pathogenesis and progression. Of interest, Mg levels are often deranged in neurodegenerative diseases and might therefore favor their onset and aggravation [34]. We briefly summarize current knowledge on the connection existing between altered Mg levels and Alzheimer’s and Parkinson’s diseases, as well as multiple sclerosis.

### 4.1. Magnesium and Alzheimer’s Disease

Alzheimer’s disease is a neurodegenerative disorder that primarily affects cognitive function and accounts for 60–70% of all dementia cases [66]. It is characterized by a diffuse atrophy of the brain, which starts from the hippocampus and spreads to the rest of the brain. While the pathogenesis of the disease is complex and, in part, unveiled, a role seems to be played by the deposition of the amyloid β (Aβ) peptide as the result of the proteolytic cleavage of the β-amyloid precursor protein (APP) through the action of α and γ-secretases [67]. On the contrary, when α and β secretases cleave APP, soluble fragments are released. In the extracellular space, aggregates of Aβ peptide form the amyloid plaques, while in neurons, abnormal precipitates of the phosphorylated tau protein accumulate as neurofibrillary tangles. Aggregated proteins activate microglia and trigger neuroinflammation, which plays a role in the severity of the disease [68]. Accordingly, high levels of pro-inflammatory cytokines are detected in the brain and cerebrospinal fluid of Alzheimer’s disease patients. Additionally, inducible nitric oxide synthase (iNOS) is overexpressed in the brain, and iNOS knock-out mice are protected against the disease [69]. Additionally, PGD2 levels are higher in the cortex of Alzheimer’s disease patients than in healthy individuals. Inflammatory alterations, together with Aβ deposits, also occur in the vessels, thus impairing blood flow. Mg deficiency contributes to Alzheimer’s disease and reduced amounts of Mg have been detected in the brain of the patients [70,71,72]. While it is likely that a BBB dysfunction might alter Mg transport, no causative mechanisms have been demonstrated yet. A recent metanalysis of 21 studies published in the last 20 years reported that circulating Mg levels in Alzheimer’s patients were significantly lower than in healthy people [73]. Considering that the content of Mg in the Western diet is low [74], it is interesting that high dietary Mg intake is inversely correlated with the risk of developing cognitive impairment [75,76]. Accordingly, Mg supplementation in an animal model of Alzheimer’s disease supports the relevance of appropriate concentrations of this mineral in reducing neuroinflammation, preventing Aβ deposition, and improving learning abilities and memory [77].

These protective effects of Mg are explained by the fact that Mg downregulates IL-1β, thereby inhibiting inflammation; it reduces the synthesis of the Aβ peptide by switching β- to α-secretase cleavage, facilitates the clearance of Aβ fibrils by controlling the permeability of the BBB, and prevents the phosphorylation of tau [77,78,79] (Figure 3).

Mg deficiency might represent the common denominator explaining, at least in part, the relationship existing between diabetes type 2 and Alzheimer’s disease [80]. It is well-known that Mg balance is impaired in diabetes and contributes to glucose intolerance and insulin resistance [81]. The resulting hyperinsulinemia and hyperglycemia increase Aβ accumulation and exacerbate neuroinflammation [80,82]. Correcting Mg might therefore represent a “two birds with one stone” intervention.

Mg homeostasis depends on the activity of several transporters. Transient receptor potential melastatin (TRPM) 7, a ubiquitously expressed ion channel coupled with an α kinase domain, is essential in maintaining intracellular Mg concentrations [83]. It has a role in neurodevelopment at early embryonic stages and coordinates the maturation of neuronal networks at later postnatal stages [84] and the loss of this channel is involved in the onset of neurological diseases [85]. At the cellular level, TRPM7 controls the proliferation and differentiation of neuronal cells and regulates the growth and migration of astrocytes [8]. TRPM7 has also been implicated in mediating neuronal death in response to various triggers [86]. Accordingly, in the rat hippocampus, TRPM7 knock-down prevents neuronal death after ischemia, thus preserving cognitive function and memory tasks [87]. Moreover, the evidence that TRPM7 shapes immune system function [88,89] indicates it might play an indirect role in neuroinflammation and neurodegeneration. Of interest, TRPM7 kinase transactivates SMAD2-dependent genes, thus promoting T cell differentiation toward the pro-inflammatory Th17 lymphocyte [90]. IL17, the signature cytokine of Th17 cells, promotes the activation of astrocytes and microglia, heightens the inflammatory cascade [91], and impairs BBB integrity [92]. All these events occur in Alzheimer’s disease, and IL17 promotes the classical Alzheimer’s disease neuropathology [92]. Circulating TH17 cells are markedly increased in the early stages of the disease and are associated with the extent of amyloidopathy [93]. Antibodies against IL17 mitigate cognitive impairment and reduce amyloid β-induced neuroinflammation in mice [94].

### 4.2. Magnesium and Parkinson’s Disease

Parkinson’s disease is the second most common neurodegenerative disorder, and its prevalence has risen rapidly in the last 20 years. It is characterized by cardinal motor postural instability, rigidity, involuntary body movements, bradikynesia, rest tremor- and non-motor symptoms, such as cognitive impairment, depression, autonomic dysfunction, and disorders of sleep [95]. Parkinson’s disease results from the complex combination of various pathological events, among which are abnormal protein aggregation, altered lysosome function, mitochondrial dysfunction, endoplasmic stress, glutaminergic excitotoxicity, oxidative stress, and neuroinflammation. These all promote neuronal death, targeting the dopaminergic neurons of the basal ganglia in particular. Mg might exert beneficial effects by contrasting oxidative stress, neuroinflammation, and exaggerated NMDA-R activity. Further, Mg increases BDNF, which is a trophic factor for dopaminergic neurons, including those that degenerate in the disease [96]. Moreover, by acting on the mTOR pathway, Mg reduces the formation of α-synuclein, the main protein involved in the pathogenesis of Parkinson’s disease [97]. Mg also inhibits α-synuclein aggregation by directly binding the protein and preventing interaction with pro-aggregant metals, such as Fe [2]. These mechanisms might explain why a multicenter case-control study in Japan reported that higher Mg intake was associated with a reduced risk of Parkinson’s disease [98]. Moreover, since nitrative stress has a high etiopathogenic relationship with the disease [99], it should be recalled that Mg downregulated iNOS and attenuated the loss of dopaminergic neurons in a murine Parkinsonian model [100].

Mg in the cerebrospinal fluid is lower in Parkinson’s disease patients and in Parkinson’s disease animal models than in controls [101,102]. Moreover, using spectroscopic methods, lower Mg levels were detected in postmortem brains from Parkinson’s disease patients compared to the controls [103]. A low Mg dietary intake over generations damages the mitochondria, the endoplasmic reticulum, ribosomes, and nuclear DNA, as well as induces the loss of the dopaminergic neurons in the substantia nigra [104]. That altered Mg balance has a role in Parkinson’s disease is further supported by the evidence that mutations of the Mg transporting protein solute carrier family 41 member 1 (SLC41A1) were reported in familial Parkinson’s disease patients [105,106], and SLC41A1 downregulation occurs in a Parkinsonian animal model [107]. Looking at TRPM7, it has been proposed as a candidate susceptibility gene for familial Parkinson’s disease [108]. TRPM7 has a fundamental role in the survival of dopaminergic human neuroblastoma SH-SY5Y cells [109], and the suppression of TRPM7 inhibits Mg content and mitochondrial function, inducing cell death [85]. On the contrary, in an experimental model of Parkinson’s disease induced by 6-hydroxy-dopamine, the total amounts of TRPM7 are increased [110], and the silencing TRPM7 is neuroprotective in pheochromocytoma PC12 cells [111].

### 4.3. Magnesium and Multiple Sclerosis

Multiple sclerosis is a common disabling disease in young adults and is manifested by a variety of signs and symptoms, including vision loss, weakness, and spasms. Hallmarks of multiple sclerosis are the demyelination and axonal degeneration in different areas of the brain and spinal cord, as well as the disruption of the BBB, events sustained by the activation of inflammation [112]. Cytokines, NO, and mitochondrial dysfunction are major determinants in the onset, recurrence, and progression of the disease. Considering the anti-inflammatory action of Mg and its role in maintaining mitochondrial function, the potential protective role of a correct intake of Mg is feasible. As in Alzheimer’s disease, Th17 cells are key effectors in multiple sclerosis, and accordingly, IL17 is highly expressed in the brain of patients [113]. Moreover, because of its antagonist action on Ca, Mg supplementation improved the severe spastic paraplegia in a young multiple sclerosis patient [114]. Further, since vitamin D deficiency seems to play a causal role in the disease, it is worth noting that Mg is required for vitamin D metabolism, and Mg supplementation is recommended in vitamin D deficiency [115,116]. Notably, patients with multiple sclerosis treated with Mg and vitamin D experienced significantly less exacerbations than untreated ones [117]. Conflicting results are available about the levels of serum Mg. A study reported that patients with normal serum Mg had a more favorable clinical course than patients with low circulating Mg [118]. Another investigation demonstrated reduced concentrations of several minerals, among which were Mg, in the blood of multiple sclerosis patients [119]. However, according to a recent metanalysis, no significant differences of magnesemia emerged between multiple sclerosis patients and healthy controls [120].

TRPM7 is overexpressed in astrocytes isolated from patients with multiple sclerosis, and this dysregulation seems to be involved in the early and late steps of multiple sclerosis. Initially, upregulated TRPM7 might drive the accumulation of Th17 lymphocytes, thus favoring inflammation [89]. Then, it might favor the progression of the disease because TRPM7 overexpression is linked to the formation of the gliotic scars [121].

## 5. Unanswered Questions in Mg, Neuroinflammation and Neurodegeneration

Some aspects that might correlate with altered Mg levels and neurodegenerative diseases still need to be unveiled. First, Mg deficiency accelerates cellular senescence and aging [122,123], events that translate to reduced adaptation to stressors, as typically reported in the elderly. Moreover, senescent cells, including astrocytes, release a wide range of inflammatory cytokines and chemokines, thus generating and maintaining inflammation with detrimental effects on the brain [124]. Looking at the increasing interest in senotherapeutic approaches to prevent or treat age-associated disorders [125], it is rational to investigate whether Mg deficiency might also promote neurodegeneration by hastening the senescence of brain cells. Second, Mg is a regulator of the circadian rhythm of cellular metabolism and function [126]. Indeed, circadian control of oxidative stress, proteostasis, and inflammatory pathways plays a role in neurodegeneration. Therefore, it is likely that alterations of Mg balance are reflected into circadian dysfunction, which results in the accumulation of neurotoxic proteins. Accordingly, circadian rhythm disruptions, which are common in the elderly, are more severe in patients with a wide range of neurodegenerative diseases, including Parkinson’s and Alzheimer’s diseases [127]. Third, Mg supports the integrity of the intestinal barrier. Mg deficiency causes intestinal dysfunction that increases the paracellular transport of LPS, a component of the cell wall of Gram-negative bacteria, into systemic circulation [128,129], thus promoting inflammation. Therefore, Mg deficiency also increases neuroinflammation by altering gut microbiota, which influences the nervous system through the activation of the vagal nerve, the production of cytokines, and the release of neuropeptides and neurotransmitters [130]. Accordingly, low Mg intake unbalances the gut microbiota–brain axis, which leads to depression in mice [130,131]. Moreover, because of the growing interest in the implications of TRPM7, a cation channel with kinase function, in neurological and neurodegenerative disorders, there is a need to better understand the role of this channel in the direct (on target cells) or indirect (e.g., via systemic inflammation) impact on nervous system alterations.

Another aspect that requires attention is the choice of Mg salt that more efficiently traverses the BBB to be utilized in case of Mg deficiency to harmonize the levels of brain Mg. While preclinical reports are rather numerous, more clinical trials should be fostered to address this issue.

A correct and, if possible, personalized dietary intake of Mg might represent a preventive measure, whereas supplementing Mg might be an adjunct option in neurodegeneration.

## 6. Conclusions

Neuroinflammation drives tissue damage in neurodegeneration. Solid evidence suggests a role of Mg in taming neuroinflammation and in retarding some neurodegenerative diseases. Therefore, a correct and, if possible, personalized dietary intake of Mg might represent a preventive measure, whereas supplementing Mg might be an adjunct option in neurodegeneration.

## Figures and Tables

**Figure 1 ijms-24-00223-f001:**
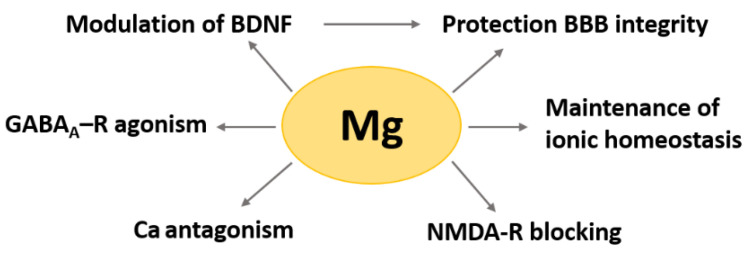
The role of magnesium in the brain. Magnesium in the brain plays a fundamental role in the modulation of different pathways.

**Figure 2 ijms-24-00223-f002:**
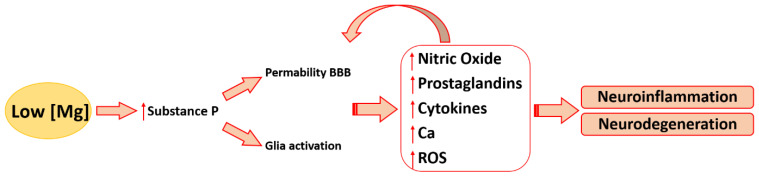
The effects of low magnesium in the brain. Magnesium deficiency is responsible for the increase of neuroinflammation and BBB dysfunction.

**Figure 3 ijms-24-00223-f003:**
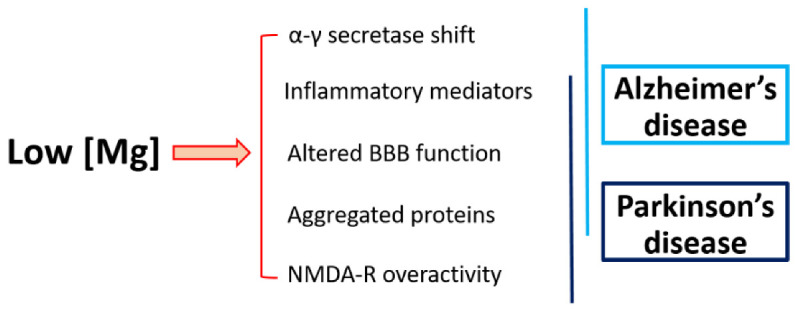
Low Mg in Alzheimer’s and Parkinson’s diseases. Magnesium deficiency promotes common events involved in Alzheimer’s and Parkinson’s diseases.

## Data Availability

Not applicable.

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
