# Peer review of "Magnesium and the Brain: A Focus on Neuroinflammation and Neurodegeneration"

_ijms, 2022, doi:10.3390/ijms24010223_

Round 1
Reviewer 1 Report
This is a well-documented up-to-date review of magnesium biochemical roles that are involved in brain function and thus likely to have an effect on neuroinflammation and neuroinflammatory diseases. I only have a couple of minor comments. First, the number of times the word "indeed" is used gets a little annoying; this tern could be reduced in the text. Other words that could be reduced are "of note," "of interest," and "notably.
Conclusions usually summarizes the previous content of the article in a succinct fashion and what they mean for the future. Thus, the material about circadian rhythms and intestinal dysfunction seems to come out of nowhere. It may be more appropriate for this to be discussed briefly somewhere in the main section of the article (magnesium and other contributors to neuroinflammation?}.
Is multiple sclerosis usually capitalized?
Author Response
We thank the reviewer for the appreciation of our work and for the comments.
- The number of times the word "indeed" is used gets a little annoying; this tern could be reduced in the text. Other words that could be reduced are "of note," "of interest," and "notably.
We thank the reviewer for this observation. We have amended the manuscript accordingly.
- Conclusions usually summarizes the previous content of the article in a succinct fashion and what they mean for the future. Thus, the material about circadian rhythms and intestinal dysfunction seems to come out of nowhere. It may be more appropriate for this to be discussed briefly somewhere in the main section of the article (magnesium and other contributors to neuroinflammation?}.
We have added paragraph 5 entitled “Unanswered questions in Mg, neuroinflammation and neurodegeneration” to discuss some topics that future research should address. Paragraph 6 “Conclusions” is now very concise and suggest that an appropriate Mg intake/supplementation might represent a useful tool to prevent neuroinflammation and neurodegeneration.
- Is multiple sclerosis usually capitalized?
Thank you for this comment. Multiple sclerosis is not capitalized and we have corrected the manuscript accordingly.

Reviewer 2 Report
It is a well-written review. I have only small suggestions for the authors. First, I am not sure whether we should use the phrase "Mg homeostasis" in the case when the organismal level of MG is not actively controlled. Second, can you accept the notion that Mg anty inflammatory effect is due to the limitation of Ca2+ activity at the synaptic and intraneuronal levels? Mg simply limits the Ca2+ inflows to neural cells and this effect may inhibit apoptosis.
Author Response
We thank the reviewer for the appreciation of our work and for the comments.
- I am not sure whether we should use the phrase "Mg homeostasis" in the case when the organismal level of MG is not actively controlled.
We thank the reviewer for this comment. We have amended the manuscript.
- Can you accept the notion that Mg anty inflammatory effect is due to the limitation of Ca2+ activity at the synaptic and intraneuronal levels? Mg simply limits the Ca2+ inflows to neural cells and this effect may inhibit apoptosis.
In this paper we have not mentioned the complex and controversial role of Mg in apoptosis, since our aim was to link Mg, neuroinflammation and neurodegeneration. The issue is very interesting and deserves a deeper critical review of the available literature.
